# Active Participatory Regional Surveillance for Notifiable Swine Pathogens

**DOI:** 10.3390/ani14020233

**Published:** 2024-01-11

**Authors:** Giovani Trevisan, Paul Morris, Gustavo S. Silva, Pormate Nakkirt, Chong Wang, Rodger Main, Jeffrey Zimmerman

**Affiliations:** 1Department of Veterinary Diagnostic and Production Animal Medicine, College of Veterinary Medicine, Iowa State University, Patterson Hall, 1800 Christensen Drive, Ames, IA 50011-1134, USA; trevisan@iastate.edu (G.T.); gustavos@iastate.edu (G.S.S.); chwang@iastate.edu (C.W.); rmain@iastate.edu (R.M.); 2Department of Statistics, College of Liberal Arts and Sciences, Iowa State University, Snedecor Hall, 2438 Osborn Drive, Ames, IA 50011-4009, USA; psmorris@iastate.edu (P.M.); pnakk@iastate.edu (P.N.)

**Keywords:** surveillance, participatory surveillance, swine, early detection, targeted sampling

## Abstract

**Simple Summary:**

Effective, sustainable regional surveillance for the early detection of notifiable swine pathogens has been difficult to achieve. Regional surveillance based on clinical signs (syndromic surveillance) is not diagnostically sensitive and specific. Surveillance based on farm-by-farm testing is burdensome and costly. Borrowing the strengths of each approach, we evaluated an active participatory surveillance design in which regional status was determined by targeted sampling of 10 poor-doing pigs in each participating farm followed by screening in credentialed laboratories. The analysis showed that at 0.1% prevalence (18 infected farms among 17,521 farms) and a farm-level detection probability of 30%, active participatory surveillance would detect ≥ 1 positive farms with 67%, 90%, and 97% probability when producer participation was 20%, 40%, and 60%, respectively. Depending on the specimen collected (serum or swab sample) and test format (nucleic acid or antibody detection), the cost per round of sampling ranged from EUR 0.016 to EUR 0.032 (USD 0.017 to USD 0.034 USD) per pig in the region. The techniques and technologies required for active participatory surveillance are widely available and in common use. Implementation would require coordination among producers, industry groups, and animal health authorities.

**Abstract:**

We evaluated an active participatory design for the regional surveillance of notifiable swine pathogens based on testing 10 samples collected by farm personnel in each participating farm. To evaluate the performance of the design, public domain software was used to simulate the introduction and spread of a pathogen among 17,521 farms in a geographic region of 1,615,246 km^2^. Using the simulated pathogen spread data, the probability of detecting ≥ 1 positive farms in the region was estimated as a function of the percent of participating farms (20%, 40%, 60%, 80%, 100%), farm-level detection probability (10%, 20%, 30%, 40%, 50%), and regional farm-level prevalence. At 0.1% prevalence (18 positive farms among 17,521 farms) and a farm-level detection probability of 30%, the participatory surveillance design achieved 67%, 90%, and 97% probability of detecting ≥ 1 positive farms in the region when producer participation was 20%, 40%, and 60%, respectively. The cost analysis assumed that 10 individual pig samples per farm would be pooled into 2 samples (5 pigs each) for testing. Depending on the specimen collected (serum or swab sample) and test format (nucleic acid or antibody detection), the cost per round of sampling ranged from EUR 0.017 to EUR 0.032 (USD 0.017 to USD 0.034) per pig in the region. Thus, the analysis suggested that an active regional participatory surveillance design could achieve detection at low prevalence and at a sustainable cost.

## 1. Introduction

In this study, a swine farm is defined as a specific geographic location where a population of pigs under one management system is raised; a region is defined as a contiguous geographical area within which the farms under surveillance are located. Swine farms are diverse in size and structure, production type, housing, and management, but the trend over the last few decades has been toward fewer and larger farms. As an example, the number of U.S. farms with pigs declined from 168,450 in 1995 to 68,300 in 2012 [1] while the average farm inventory increased from 302 pigs to 1044 [2]. This period also saw the emergence of specialized swine farms and widespread adoption of the practice of moving young pigs from breeding-specific farms to feeding-specific farms. Thus, in 2019, Denmark, France, Germany, and Spain cumulatively imported 15.7 million and exported 20.5 million live pigs [3] and, in the U.S., 63.4 million live pigs were transported from one state to finishing farms in other states [4]. Overall, these changes have contributed to improved production efficiency but complicated disease control. Logically, the greater movement of animals, personnel, and material facilitates the spread of infectious agents. For example, porcine epidemic diarrhea virus spread to at least 12 states within 8 weeks of its initial detection in the U.S. [5,6]. 

Under these circumstances, the early detection of notifiable swine pathogens is essential but difficult. In Brazil (1978), an unrecognized outbreak of African swine fever virus (ASFV) in the index farm was followed by its spread to 11 states. Eradication took 8 years and cost ~USD 20 million [7]. In a 1997–1998 outbreak in the Netherlands, a retrospective analysis determined that classical swine fever virus (CSFV) was spreading in the country 5 to 7 weeks prior to its recognition [8]. Eradication was ultimately accomplished at a cost of ~USD 2.3 billion [9]. In the United Kingdom (2001), foot-and-mouth disease virus (FMDV) infections went unnoted and the virus was disseminated widely via the movement of infected animals. Eradication took 6 months, led to the euthanasia of 4 million animals, and cost ~USD 4.0 billion [10]. At present, the ASFV pandemic initiated in 2007 continues to expand despite the recognition that “*an early detection system for ASF could facilitate early reporting and response (and limit) the spread of the disease*” [11].

The need for effective, on-going regional surveillance is obvious, but a workable design is not [12]. Surveillance based on “down-the-road” testing to prove farms free from infection is often performed in government-supported eradication programs, e.g., Aujeszky’s disease [13], but is costly and administratively burdensome. Syndromic surveillance [14], i.e., detection based on reports of clinical signs consistent with the pathogen(s) of interest, should meet the need, but Poppensiek and Budd, cited in [15], found that “*The greatest single difficulty in a disease-reporting program proved to be the failure of vets to file reports*”. Exploring this problem, Gates et al. [16] found that the reluctance to report arose from feelings of uncertainty, fear of the consequences of reporting, distrust of authorities, and unfamiliarity with the reporting process. Participatory surveillance, i.e., including members of the population at risk in the surveillance data collection process [17,18,19,20], has improved syndromic surveillance, but its effectiveness is limited by the participants’ clinical experience and the inherent diagnostic ambiguity of clinical signs. Thus, Elbers et al. [21] estimated that a diagnosis of CSFV based solely on clinical signs achieved a diagnostic sensitivity of 73% and a diagnostic specificity of 53%. 

The objective of this study was to characterize the performance of a surveillance design best described as “collecting and testing a few targeted samples from each of many farms in the region”. In more formal terms, we analyzed the performance and cost of test-based, regional, active participatory surveillance based on the targeted sampling of 10 poor-doing pigs by farm personnel (producer, staff, and/or veterinarian) followed by screening for the pathogen of interest in credentialed laboratories. Because our objective was to explore the feasibility and performance of this general design, the analysis did not include a sampling and testing process for a specific pathogen. However, a key assumption in the cost analysis was that 10 pigs would be sampled on each participating farm and the individual pig samples combined into two pools (5 pigs per pool) for antibody or nucleic acid testing. 

## 2. Study Design

The study was conducted in three phases. In Phase 1, the spread of a notifiable (but unspecified) pathogen was simulated in a population of 17,521 swine farms holding 51,515,699 pigs in a geographic region of 1,615,246 km^2^ for a period of 70 days. Based on the simulated farm status (negative or positive), Phase 2 estimated the probability of detecting ≥ 1 positive farms in the region as a function of farm-level sensitivity (10%, 20%, 30%, 40%, 50%), percent of farms participating in the surveillance program (20%, 40%, 60%, 80%, 100%), and farm-level prevalence in the region. Since the objective was to broadly evaluate the performance of the surveillance design, Phase 1 (pathogen spread) and Phase 2 (detection) simulations were performed over a range of parameters. In Phase 3, active participatory surveillance was analyzed in terms of the cost per farm and the cost per pig in inventory per round of sampling and testing.

### 2.1. Phase 1: Simulating Pathogen Spread—Animal Disease Spread Model (ADSM)

The Animal Disease Spread Model (ADSM) is public-domain software designed to simulate the spread of infectious agents in livestock populations [22]. ADSM uses a static, fixed population and defines the population of animals at a single geographic location, i.e., a farm, as the epidemiological unit. For Phase 1 simulations, a population of swine farms was created from publicly available concentrated animal feeding operation (CAFO) permit data provided by the appropriate authorities in the states of Colorado, Iowa, Kansas, Minnesota, Missouri, Nebraska, Oklahoma, and South Dakota. State datasets were collated into a single ADSM-compatible file. Farms determined to be inactive or with data quality issues were removed, resulting in a final data set consisting of 17,521 farms (Table 1). Swine packing plant locations and slaughter capacities were included in the population file to account for their role in indirect pathogen spread [23], with latitude and longitude generated from their addresses [24] using Google Maps (www.google.com/maps accessed 1 April 2021). 

ADSM (version 3.510.0) software required the identification of each farm site by production type (breeder, feeder, or breeder/feeder), inventory (number of animals), and geolocation (latitude and longitude). For the majority of farms, production type was provided in the state datasets or derived from the site name, e.g., “Smith Sow Farm”. Using this approach, 13,041 of 17,521 farms in the population file were assigned to production type: 209 (1.6%) breeder/feeder, 702 (5.4%) breeder, and 12,130 (93.0%) feeder. The remaining 4481 farms were randomly assigned [25] to production type proportional to state-level production types or, if state data were not adequately reported, the overall proportions in the database. State-level permit data described the capacity (inventory) of each farm either as the number of pigs or as “animal units”. In the latter case, animal units were converted to the number of pigs on the basis of one animal unit per 2.5 pigs weighing ≥ 24.9 kg (≥55 pounds) or 10 pigs weighing < 24.9 kg (<55 pounds) [26]. With one exception, all states reported farm location by latitude and longitude, by ZIP Code (i.e., postal code), or by county (i.e., an administrative subdivision of a state). For farms without precise geolocation, the spsample function in the sp R package (version 1.4-5) [27] was used to randomly generate a latitude and longitude within the geographic unit associated with the record (i.e., ZIP Code, county, or state).

#### 2.1.1. ADSM Simulations

The ADSM software was designed to simulate the spread of a designated pathogen in a defined livestock population by setting parameter values representative of the pathogen’s transmission characteristics and industry production practices. In this study, ADSM simulations were performed over a range of parameter values (Table 2) to provide spread estimates generalizable to a variety of notifiable pathogens. Although disease control options are available in ADSM, e.g., movement restrictions, vaccination, and farm depopulation, they were not implemented so as to allow the unrestricted spread of the hypothetical agent within the region. 

For simplicity, each spread scenario began with a single index farm (Table 2). A total of 30 index farms were identified by randomly selecting 10 farms from each of the 3 pig density categories (1.1–3.3 pigs per km^2^, 15.9–25.3 pigs per km^2^, 106.8–214.5 pigs per km^2^) using features built into R (version 4.1.0) software [25]. Each of these 30 individual index farms was successively categorized as a breeder, breeder–feeder, or feeder in simulations. This process ensured that the spread scenarios covered the range of possible outcomes that could arise due to differences in pig density in the region surrounding the index farm and in the direct and indirect contact rates among production types. A total of 2430 pathogen spread scenarios were simulated based on all combinations of county-level pig density (*n* = 3), index farm within county density (10 farms in each of the 3 county-level pig densities), index farm production type (*n* = 3, i.e., breeder, feeder, breeder–feeder), spread by direct contact (*n* = 3, i.e., probability levels 0.2, 0.4, 0.6), spread by indirect contact (*n* = 3, i.e., probability levels 0.05, 0.10, 0.15), and area spread (*n* = 3, i.e., probability levels 0.001, 0.010, 0.100). Each scenario was replicated 100 times to account for the stochastic nature of the ADSM simulations.

#### 2.1.2. ADSM Automation

After constructing the initial pathogen spread scenario using the ADSM Scenario Creator, the creation of the subsequent 2429 scenarios was automated. In brief, the Scenario Creator output a directory that contained an SQLite database (“ScenarioX.db”) into the ADSM workspace that housed the information ADSM used to run the scenario. An R script [25] was written to copy the database file and update the unique parameter values for each scenario (Table 2), thereby creating additional SQLite databases suitable to be imported and run on ADSM. 

The procedure was performed as follows:A new file directory in the ADSM workspace was created using the base R function dir.create and the saved template scenario “ScenarioX.db” was copied from the initial simulation. This file was renamed using the base R function file.rename, e.g., “new_scenario.db”.A connection was created between R and the SQLite database using the dbConnect function from the RSQLite R package (version 2.2.4) [30] to access the “new_scenario.db” file for editing, (i.e., con = dbConnect(SQLite(), dbname = “new_scenario.db”)).Once the connection was opened, the dbGetQuery function was used to bring the SQLite table to be edited into the R environment as a data frame. For example, the R code created a data frame in the R environment named “Population” using the ScenarioCreator_unit table from the SQLite database. This table contained the entire population file input during the ADSM scenario creation process.a.Population <- dbGetQuery(con, “SELECT * FROM ScenarioCreator_unit”).b.Other tables altered using this procedure included those containing the direct spread parameters (ScenarioCreator_directspread), indirect spread parameters (ScenarioCreator_indirectspread), and local area spread parameters (ScenarioCreator_airbornespread).R functions were then used to update the data frame to fit the new desired scenario (i.e., production types, initial infection statuses, or transmission probabilities).The dbWriteTable function with the overwrite option specified as TRUE was used to replace the SQLite table in the database file with the newly edited data frame. For example, dbWriteTable(con, name = “ScenarioCreator_unit”, value = Population, overwrite = TRUE).Rerunning the dbConnect line exactly as written in Step 2 saved the SQLite database file with the changes included (i.e., con = dbConnect(SQLite(), dbname = “new_scenarioX.db”)).

At the conclusion of the scenario creation process, the spread scenarios were run using batch processing (see https://github.com/NAVADMC/ADSM/wiki/Batch-processing-of-scenarios-using-ADSM-Auto-Scenario-Runner accessed on 1 May 2021).

#### 2.1.3. Phase 1: Spread Results

Phase 1 simulation results (100 iterations for each of the 2430 scenarios) are reported in Table 3 as the mean number of infected farms on simulation day 70 for all possible combinations of the spread parameter values listed in Table 2, i.e., index farm county pig density (*n* = 3 pig densities), index farm type (*n* = 3), probability of transmission by direct contact (*n* = 3 levels), indirect contact (*n* = 3 levels), and area spread (*n* = 3 levels). All parameters in the model affected the outcome, but holding all other parameters constant (ceteris paribus), it can be seen that the probability of transmission by direct contact, i.e., the movement of infectious animals among sites, was the most impactful in terms of the total number of infected farms on day 70.

### 2.2. Phase 2: Simulating Pathogen Detection

Among the 2430 spread scenarios simulated in Phase 1, the 360 scenarios indicated in Table 4 were selected for use in Phase 2. Each of the 12 groups of spread scenarios shown in Table 4 consisted of 30 scenarios, i.e., 10 index farms in each pig density category, with each index farm successively classified as one of 3 production types (breeder, breeder–feeder, and feeder). For each of these 360 spread scenarios, the detection of ≥ 1 positive farms in the region was simulated under 25 pathogen detection settings based on farm-level sensitivity (10%, 20%, 30%, 40%, or 50%) and farm participation in the surveillance program (20%, 40%, 60%, 80%, or 100% of farms in the region). For the 20%, 40%, 60%, and 80% participation levels, farm participation was allocated uniformly across the 3 pig inventory size categories in Table 1 through simple random sampling without replacement. For example, simulations at 20% participation included 20% of the farms in the ≤1000 category, 20% in the 1001–4999 category, and 20% in the ≥5000 category. For each participation level, 1000 farm groupings were randomly selected using R software [25] to match the 1000 surveillance simulations. By definition, 100% participation did not require participant selection.

An R function [25] was written to perform the surveillance simulations for each combination of spread scenario replicate and detection setting, as described below. Because each spread scenario was replicated 100 times, there were a total of 100,000 iterations (1000 iterations for each of the 100 replicates) for each combination of spread scenario and detection setting.

For iteration i, where i = 1, …, 1000:Assign as participants the ith set of participating farms from the list of pre-selected sets corresponding to the current setting’s participation level. For 100% participation, the entire population of farms were participants.For each participating farm, the farm infection status (negative, positive) was identified for days 7, 14, 21, 28, 35, 42, 49, 63, and 70 of the ADSM spread simulation.For each of the days listed in Step 2, participating farms classified as positive were “tested” (simulated) independently in R using the rbinom function with the probability of detection equaling the assigned farm-level sensitivity. Thus, for each positive farm and where p was the assigned farm-level sensitivity, rbinom (*n* = 1, size = 1, prob = p) randomly generated a 0 or 1, where 1 indicated that the infection was detected.

#### 2.2.1. Phase 2: Detection Results

For each of the 25 detection settings, results were reported as the probability of detection by regional farm-level prevalence, with the probability of detection calculated as the percentage of iterations in which ≥ 1 true positive farms “tested” positive in the simulations. Results are provided in Table 5 and Figure 1 and Figure 2 by regional prevalence, farm-level sensitivity, and producer participation. The analysis showed that detection was dependent on the interactions between producer participation and farm-level sensitivity, but high probabilities of detection were achieved at low prevalence over a wide range of participation and sensitivity values. For example, at 0.1% prevalence (18 positive farms among 17,521 farms) and a farm-level detection probability of 30%, the participatory surveillance design achieved 67%, 90%, and 97% probabilities of detecting ≥ 1 positive farms in the region when producer participation was 20%, 40%, and 60%, respectively.

### 2.3. Phase 3: Cost of Sampling and Testing

The cost analysis assumed that ante mortem specimens (blood, blood swabs, nasal swabs, oral swabs, or fecal swabs) would be collected from 10 poor-doing pigs in each participating farm, combined into 2 pooled samples (5 pigs per pool), shipped to a credentialed laboratory in an insulated shipping container with coolant, and tested by polymerase chain reaction (PCR) or antibody ELISA. Since the general surveillance design did not define a specific testing protocol, costs were estimated for 3 cases: serum samples tested by PCR, swab samples tested by PCR, and serum samples tested by ELISA. To further evaluate the impact of testing costs on overall program costs, 3 price levels for PCR (EUR 18.65, EUR 23.32, EUR 27.98/USD 20.00, USD 25.00, USD 30.00) and antibody ELISA (EUR 4.66, EUR 7.00, EUR 9.33/USD 5.00, USD 7.50, USD 10.00) were used in the estimates.

The estimated cost of a single round of sampling used the inputs and costs listed in Table 6 and assumed 100% producer participation. Costs listed in Table 6 are the mean of prices quoted by 3 companies for the distribution of products in the U.S. Supplies for collecting serum samples included single-use blood collection tubes and needles (*n* = 10), tubes in which to pool samples (*n* = 2), and disposable gloves (2 pairs). Supplies for swab samples included swabs (*n* = 10), transport medium, tubes in which to pool samples (*n* = 2), and disposable gloves (2 pairs). Package shipment costs reflect rates paid by clients of the Iowa State University Veterinary Diagnostic Laboratory (Ames, IA, USA) and may vary.

The analysis assumed that the labor and materials required to collect, process, and package samples for shipment would be provided by the farm and, therefore, were not included in the cost analysis. Likewise, it was assumed that blood samples would be centrifuged and the serum pooled (5 pigs per pool) prior to shipment in order to avoid processing charges at the laboratory. Some costs that would be expected in the normal course of sampling and testing were also not included. For example, no attempt was made to account for the added cost of duplicate sampling or testing, e.g., the cost of an additional tube and needle for a second attempt at blood collection from a pig or the cost of retesting a non-negative sample in the laboratory. Likewise, the cost analysis did not include the costs required to administer and coordinate the program.

#### 2.3.1. Phase 3: Estimated Cost of Sampling and Testing

The results of the cost analysis are listed in Table 7 for 3 “specimen by test” combinations with 3 costs for each test. The estimates are given in terms of the average cost per farm in the region, the average cost per pig in the region, and the average cost per pig in inventory for the farm size categories given in Table 1, i.e., farms with ≤1000 pigs, 1001–4999 pigs, and ≥5000 pigs. Using a PCR cost of EUR 23.32 (USD 25.00) or ELISA cost of EUR 7.00 (USD 7.50) per sample, the cost of sampling and testing would be approximately EUR 0.03 (USD 0.03) or EUR 0.02 (USD 0.02) per pig in the region, respectively. On a farm basis, given that sample size and test costs are the same for all farms, the cost per pig increases as the farm pig inventory decreases, as shown in Table 7.

## 3. Discussion

Concerning surveillance systems, Thacker et al. [31] advised that “*Simplicity should be a guiding principle …. Simple systems are easy to understand and implement, cost less than complex systems, and provide flexibility*”. Consistent with the theme of “simplicity”, the active participatory regional surveillance design was based on targeted sampling of 10 live but poor-doing pigs on participating farms by farm personnel, followed by testing in credentialed laboratories. The design differed most from traditional surveillance in that it focused on the status of the region rather than the status of individual farms. The result was fewer samples per farm yet sensitive regional surveillance at a manageable cost (Table 5 and Table 7). 

Targeted sampling, already recommended for the surveillance of CSFV [32] and ASFV [33], addressed the problem of detection in populations characterized by heterogeneity and low prevalence [34]. That is, commercial swine farms separate animals into barns and pens by age, stage, and function, i.e., conditions that are inconsistent with the independence and homogeneity assumptions underlying the traditional power formula based on simple random sampling. Thus, Crauwels et al. [35] reported that random sampling would be unlikely to include a CSFV-positive pig for several weeks following its introduction into a naïve farm; that is, until it had spread sufficiently and infected a sufficient proportion of the population. 

The surveillance design called for sampling live, poor-doing pigs because notifiable pathogens may not produce remarkable clinical signs and early mortalities, including CSFV [36], ASFV [37], and FMDV [38]. Kirkland et al. [36] cautioned that CSFV strains of low or moderate virulence could circulate without notable clinical signs for 4–8 weeks. Schulz et al. [39] concluded that, depending on the virulence of the isolate, it could take up to a month for ASFV-related mortalities to be noted. Thus, sampling poor-doing live pigs would facilitate early detection by eliminating the expectation of telltale clinical signs and/or conspicuous mortalities. 

Sampling live pigs also anticipates the need to quickly resolve ambiguous (“non-negative”) test results. The typical response to a non-negative surveillance sample result is retesting the original sample using the original test or a confirmatory assay. If the retest result is conclusive, the question is resolved. If not, the fact that the samples originated from live pigs means that it is likely possible to re-sample and retest the original pigs and/or their penmates to quickly resolve the issue. On the other hand, if the samples originated from dead pigs or pigs no longer on the farm, the resolution will require extensive sampling of animals on the farm of origin and on other epidemiologically relevant farms.

In this “generic” surveillance scenario it was not necessary to designate specific specimens to be collected or tests to be performed. While blood and serum are traditional surveillance specimens, a variety of more easily collected antemortem specimens are increasingly used in diagnostics and surveillance, e.g., blood swabs, nasal swabs, oropharyngeal swabs, and rectal swabs [40]. Trevisan et al. [41] documented this trend for the period 2007 to 2018 in a study of 547,873 diagnostic cases submitted to 4 Midwestern U.S. veterinary diagnostic laboratories for porcine reproductive and respiratory syndrome virus (PRRSV) testing. In 2007, 51% of the diagnostic cases included serum samples; in 2018, 21% of cases included serum samples, 35% included oral fluid samples, and 11% included processing fluid samples. 

Sample collection by farm personnel working under the supervision of the farm veterinarian is common practice in many parts of the world. The use of easily collected antemortem samples will facilitate producer participation and is consistent with sampling by lay personnel. Regardless of the specimen(s) selected for use, the ability of lay participants to collect diagnostic samples is supported by the literature. In human medicine, Branson [42] reported that 156,121 (94.5%) of 165,194 self-collected dried blood spot specimens were acceptable for human immunodeficiency virus testing. The remaining 5.5% were disqualified for insufficient quantity, contamination, or excessive time between sampling and submission. Similarly, Tsang et al. [43] found no loss in diagnostic accuracy with self-collected oronasal swabs or oral fluid samples in a systematic review of 23 refereed studies involving severe acute respiratory syndrome coronavirus 2 (SARS-CoV-2) testing. In the veterinary literature, formal examples of the use of producer-collected samples include field studies on CSFV [44], PRRSV [45], and antibiotic resistance in *Escherichia coli* [46]. 

The use of farm personnel in sample collection acknowledges the fact that those who work with the pigs are also those most aware of recent changes in pig health and are best qualified to identify the appropriate animals to sample. Relevant to program sustainability, the use of farm personnel reduces sampling costs by eliminating the need to employ program samplers and integrates both scalability and responsiveness into the design. That is, because farm personnel are already on the farms, sample size and/or frequency can be quickly adjusted in response to changing circumstances, e.g., increased after the initial detection of the target to improve case finding and decreased after the threat is contained to reduce costs. 

The final point in the surveillance design is the testing of samples in credentialed diagnostic laboratories. “Credentialed”, in this case, refers to laboratories operating under national or international standards, e.g., ISO/IEC 17025 [47,48]. Such laboratories have operational quality management systems, proper equipment, and the technical expertise to reliably perform testing. Further, many of these laboratories are equipped with laboratory information management systems and the capacity to report test results electronically, thereby facilitating timely reporting to participants and, if needed, animal health authorities. Alternatively, testing in the field using point-of-care test devices is sometimes suggested as a means to expedite the discovery of notifiable pathogens. This may be possible in the future, but not at present. Hobbs et al. [49] reported the most fundamental problem: “*Inadequate regulatory guidance and poor industry oversight has led to a proliferation of point-of-care tests of varying quality and fitness for purpose …”*. A number of issues would need to be addressed if point-of-care tests are to be used for notifiable pathogens, but at a minimum, an accounting system for tracking kits and test results will need to be in place to avoid misuse.

The performance analyses of the active participatory regional surveillance design were based on simulations of the spread (Phase 1) and detection (Phase 2) of an unspecified pathogen in a population of naïve swine farms representative of the Midwest U.S. The swine farm dataset was created using concentrated animal feeding operation (CAFO) permit records from eight U.S. states. CAFO permitting requirements and data quality were not standardized across states, but the data required for the pathogen spread simulations (farm geolocation, inventory, and production type) were provided in most cases. After resolving inconsistencies and missing data (Section 2.1), the dataset consisted of 17,521 swine farms holding 51,515,699 pigs in a region of ~1,582,000 km^2^ [50]. As a point of reference, the geographic area of Belgium, France, Germany, Luxembourg, the Netherlands, Portugal, and Spain is ~1,584,000 km^2^ [51]. 

The purpose of Phase 1 was to create datasets of swine farms of known infection status (negative, positive) for use in the detection simulations (Phase 2). The Animal Disease Spread Model (ADSM) [15] software used in Phase 1 provided substantial modeling flexibility and has previously been used to simulate the spread of ASFV in Vietnam [49], Aujeszky’s disease virus in Thailand [52], CSFV in the Republic of Serbia [53], and PRRSV in both Uganda [54] and Canada [28]. A total of 2430 pathogen spread scenarios were simulated (Section 2.1.1) based on combinations of county-level pig densities, index herd production types, and the probabilities of transmission by direct contact, indirect contact, and area spread. Among these scenarios, 360 spread scenarios representing the range of outcomes were selected for use in the Phase 2 detection simulations. 

The objective of Phase 2 was to estimate the probability of detecting ≥ 1 positive farms in the region as a function of farm-level sensitivity, percent of farms participating in the surveillance program, and regional farm-level prevalence. The results were expressed in terms of the probability of detection by regional herd prevalence (from Phase 1 simulation results) rather than time-to-detection because of the diversity of spread rates simulated in the ADSM software. A difficulty when applying targeted sampling to surveillance is the absence of agreed-upon methods for calculating sample size and associated farm-level sensitivities. Using a modeling approach, Nielsen et al. [32,33] reported that targeted sampling of 5 sick or dead pigs in a population of 1000 pigs would detect CSFV 4 to 37 days and ASFV 13 days post-introduction with 95% probability. In the present study, a sample size of 10 pigs was considered a practical number for on-farm collection. The present study was not pathogen-specific and, in the absence of citable estimates, a conservative range of farm-level detection sensitivities (10%, 20%, 30%, 40%, and 50%) was used in the Phase 2 detection analysis. Similarly, data to inform the level of farm participation in this voluntary regional surveillance program were lacking. Consequently, adopting an approach that would inform administrators if such a program were to be initiated, a range of participation levels (20%, 40%, 60%, 80%, 100%) were evaluated for their effect on detection by prevalence. 

In Phase 3, the regional surveillance design was analyzed for the cost per farm (17,521 farms) and per pig in the region (51,515,700 pigs) for one round of sampling. Options evaluated in the analysis included specimen (serum vs. swab samples), assay format (PCR or ELISA), and 3 assay cost options. The analysis was based on the present costs of materials for collecting, shipping, and testing in credentialed laboratories (Table 6). The cost analysis assumed that the labor required to collect and package samples would be provided by participant swine producers and that administrative costs would be borne by existent animal health agencies. Using test costs of EUR 23.32 (USD 25.00) per PCR and EUR 7.00 (USD 7.50) per ELISA, the cost per farm in the region ranged from EUR 51.25 (USD 54.94) for serum tested by ELISA to EUR 83.89 (USD 89.94) for serum tested by PCR. The cost per pig in the region ranged from EUR 0.018 (USD 0.019) to EUR 0.029 (USD 0.031) for the same scenarios. 

Lee et al. [55] reported that U.S. swine producers would be willing to pay USD 0.581 (EUR 0.542) per pig per year to reduce the risk of losses from notifiable pathogens. While Lee et al. [55] focused on biosecurity, the regional surveillance design described herein would facilitate early detection and elimination, provide evidence of freedom from disease, and support access to international markets. Thus, the net effect is the amelioration of the major economic losses expected after the introduction of a notifiable pathogen at a price close to the producers’ cost constraints. 

The dataset of 17,521 farms used in this study was assembled from CAFO permits and, therefore, may be considered representative of the region. From Table 1, it can be seen that the 4422 farms (25.2%) in the smallest farm category (≤1000 pigs) held 2.67% of the pigs in the region. The 11,261 farms (71.8%) in the mid-size category (1001 to 4999 pigs) held an additional 64.3% of the pigs in the region. By definition, smaller farms have fewer pigs and, therefore, surveillance cost per pig is higher (Table 7). To be successful, participatory surveillance requires broad engagement. While it may be possible to further reduce costs associated with sampling, transport, and testing, it would be prudent to explore the means to incentivize small producer participation.

## 4. Conclusions

The regional active participatory surveillance design evaluated in this study is simple and adaptable to the surveillance of a variety of pathogens, farm animal species, or regions. Simplicity and clarity in sampling, testing, and reporting are central to the success of a participatory program because voluntary programs depend on the full confidence of the participants. In truth, there is little innovation in the proposed surveillance framework; the personnel, testing, and reporting systems are largely in place. The only possible novelty is the aggregation and interpretation of surveillance testing data at the regional level rather than the farm level. As was shown in the evaluation, this change in focus achieved highly sensitive regional detection at a low cost.

## Figures and Tables

**Figure 1 animals-14-00233-f001:**
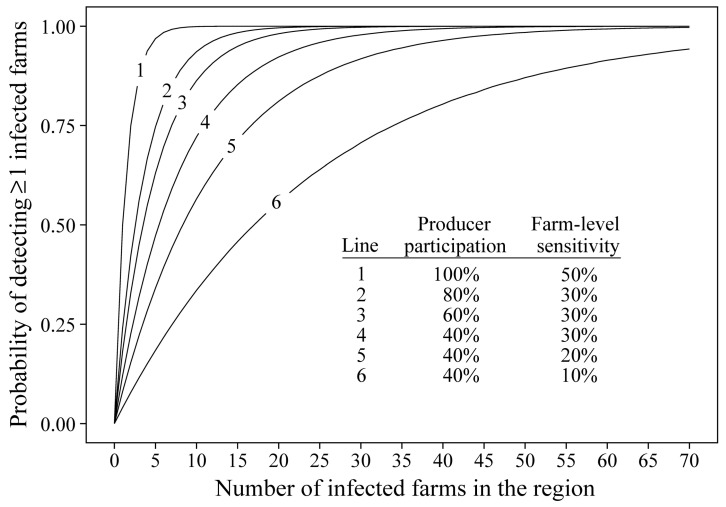
Illustration of the interaction between producer participation, farm-level detection sensitivity, and number of positive farms on the probability of detecting ≥ 1 positive farms in the region.

**Figure 2 animals-14-00233-f002:**
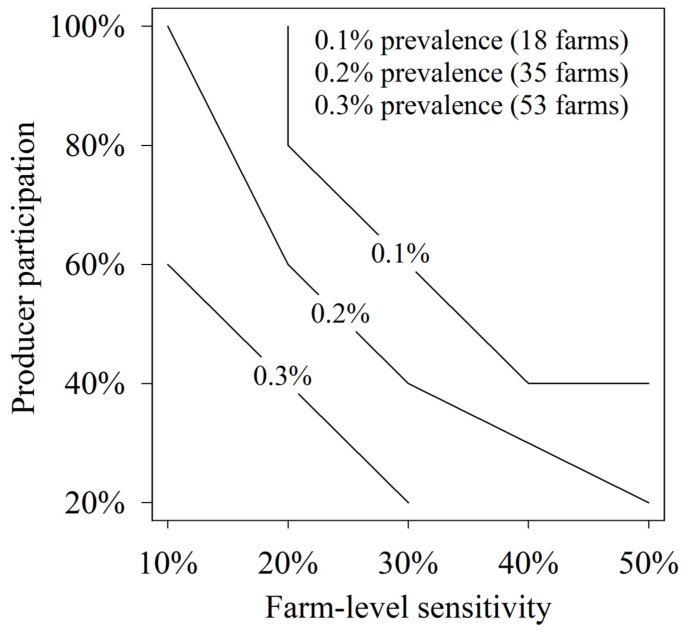
As shown for 3 prevalence levels, various combinations of producer participation and farm-level sensitivity produced ≥ 95% probability of detecting ≥ 1 positive herds in the region. Farm-level sensitivity is the probability of a positive test on samples from an infected farm.

**Table 1 animals-14-00233-t001:** Population of farm sites used in Phase 1 (pathogen spread) and Phase 2 (probability of detection) simulations by production type and pig inventory ^1^.

Total Pig Inventory	Breeder Sites (No. Pigs)	Breeder–Feeder Sites (No. Pigs)	Feeder Sites (No. Pigs)	Total Sites (No. Pigs)
≤1000	476	650	3296	4422
	(140,823)	(254,214)	(980,356)	(1,375,393)
1001 to 4999	474	294	10,493	11,261
	(1,244,232)	(641,930)	(29,374,884)	(31,261,046)
≥5000	132	120	1586	1838
	(1,556,655)	(1,595,964)	(15,726,642)	(18,879,261)
TOTAL	1082	1064	15,375	17,521
	(2,941,710)	(2,492,108)	(46,081,882)	(51,515,700)

^1^ Swine farm data based on publicly available animal feeding operation permit data provided by the appropriate authorities in the U.S. states of Colorado, Iowa, Kansas, Minnesota, Missouri, Nebraska, Oklahoma, and South Dakota.

**Table 2 animals-14-00233-t002:** Phase 1 (pathogen spread): parameters used in simulating the spread of a notifiable swine pathogen in a defined region ^1^.

Spread Parameters	Parameter Definitions and/or Values			
Index farm. a. Location (pig density). b. Production type.	First positive farm in each simulation. a. County-level pig density: low (1.1–3.3 pigs per km^2^), medium (15.9–25.3 pigs per km^2^), high (106.8–214.5 pigs per km^2^). b. Breeder, breeder–feeder, or feeder.			
2. Direct contact. a. Distance for direct contact. b. Daily movement rate.	2. Transmission by moving infectious animals among sites. a. BETAPert distribution, min 0.5 km, mode 100 km, max 1000 km. b. Fixed rate as specified by farm type:			
		Destination		
	Source farm	Breeder [28]	Feeder [28]	Packing Plant [29]
	Breeder–Feeder	NA	0.0204	0.0310
	Breeder	0.0014	0.0687	0.0310
	Feeder	NA	0.0348	0.0310
c. Probability of infecting a negative farm.	c. Probabilities tested 0.2, 0.4, and 0.6.			
3. Indirect contact. a. Distance for indirect contact. b. Daily indirect contact rate.	3. Transmission by movement of people, fomites, etc. a. BETAPert distribution, min 0.5 km, mode 100 km, max 1000 km. b. Fixed rate as specified by farm type:			
		- - - - - - - - - Destination - - - - - - - - -		
	Source farm	Breeder	Feeder	Packing Plant
	Breeder–Feeder	NA	0.0204	0.0310
	Breeder	0.0014	0.0687	0.0310
	Feeder	NA	0.0348	0.0310
c. Probability of infecting a negative farm.	c. Probabilities tested 0.05, 0.1, and 0.15.			
4. Local area spread. a. Probability of infecting a negative farm.	4. Daily probability of spread to farms ≤ 1 km from infected farm. a. Exponential drop off. Probabilities tested 0.001, 0.01, and 0.1.			

^1^ Pathogen spread simulations performed using public domain software [22] and a population of 17,521 farms (51,515,700 pigs) in a contiguous geographic region (1,615,246 km^2^).

**Table 3 animals-14-00233-t003:** Phase 1 (pathogen spread): results of the simulated regional spread of a notifiable swine pathogen reported as the mean number of infected farms on simulation day 70 for specific spread scenarios ^1^.

Index Farm Location and Type ^2^		Spread Probabilities									
		Area Spread	Direct Contact 0.2			Direct Contact 0.4			Direct Contact 0.6		
			Indirect Contact			Indirect Contact			Indirect Contact		
			0.05	0.10	0.15	0.05	0.10	0.15	0.05	0.10	0.15
Low-density county 1.1–3.3 pigs per km^2^	BF	0.001	4	4	5	16	16	21	63	67	81
	B		10	11	12	46	54	60	191	218	242
	F		6	7	8	25	31	36	99	113	138
	BF	0.010	5	5	5	19	22	25	86	96	107
	B		11	12	15	59	64	75	245	279	318
	F		7	8	10	31	39	45	136	160	182
	BF	0.100	13	14	15	85	97	119	430	427	534
	B		34	42	48	247	298	311	1127	1223	1366
	F		19	25	33	135	164	190	637	754	864
Medium-density county 15.9–25.3 pigs per km^2^	BF	0.001	4	4	4	16	17	19	64	76	79
	B		10	11	12	47	55	62	195	218	250
	F		6	7	9	27	31	37	99	126	150
	BF	0.010	5	5	6	21	23	29	88	102	112
	B		12	13	15	64	67	80	260	304	336
	F		7	10	10	35	41	49	141	164	187
	BF	0.100	16	18	18	119	120	146	456	510	595
	B		42	49	58	292	317	393	1325	1396	1539
	F		23	30	40	155	200	209	742	859	956
High-density county 106.8–214.5 pigs per km^2^	BF	0.001	5	5	5	19	20	21	66	74	85
	B		11	12	13	51	58	65	200	233	271
	F		6	8	9	26	34	41	108	132	151
	BF	0.010	6	7	8	26	31	33	109	123	141
	B		14	16	18	71	80	94	285	322	358
	F		10	11	13	42	51	55	164	182	210
	BF	0.100	47	56	67	244	286	330	982	1015	1288
	B		81	89	102	446	500	575	1661	1936	2093
	F		57	72	85	310	368	432	1239	1345	1553

^1^ Pathogen spread simulations (100 per scenario × 10 index farms) were performed using public domain software [22] in a population of 17,521 farms (51,515,699 pigs) in a contiguous geographic region (1,615,246 km^2^). ^2^ BF (breeder–feeder), B (breeder), F (feeder).

**Table 4 animals-14-00233-t004:** Spread parameter values from Phase 1 (pathogen spread) selected for use in Phase 2 (probability of detection) simulations.

Index Farm Location ^1^	Spread Probabilities									
	Area Spread	Direct Contact 0.2			Direct Contact 0.4			Direct Contact 0.6		
		Indirect Contact			Indirect Contact			Indirect Contact		
		0.05	0.10	0.15	0.05	0.10	0.15	0.05	0.10	0.15
Low-density county	0.001	-	-	-	-	-	-	-	-	-
	0.010	-	✓	-	-	✓	-	-	✓	-
	0.100	-	-	-	-	-	-	-	-	-
Medium-density county	0.001	-	-	-	-	-	-	-	-	-
	0.010	-	✓	-	-	✓	-	-	✓	-
	0.100	-	-	-	-	-	-	-	-	-
High-density county	0.001	-	-	-	-	-	-	-	-	-
	0.010	-	✓	-	-	✓	-	-	✓	-
	0.100	-	✓	-	-	✓	-	-	✓	-

^1^ Index farm located in low-density county (1.1–3.3 pigs per km^2^), medium-density county (15.9–25.3 pigs per km^2^), or high-density county (106.8–214.5 pigs per km^2^).

**Table 5 animals-14-00233-t005:** Probability of detecting ≥ 1 positive farms as a function of regional prevalence, farm-level sensitivity (%), and producer participation (%).

Regional Prevalence ^1^	Farm-Level Sensitivity (%) ^2^	Producer Participation				
		20%	40%	60%	80%	100%
0.1% (18 farms)	10	0.304	0.520	0.672	0.777	0.850
	20	0.519	0.777	0.900	0.957	0.982
	30	0.671	0.900	0.972	0.993	0.998
	40	0.776	0.956	0.993	0.999	1.000
	50	0.849	0.982	0.998	1.000	1.000
0.2% (35 farms)	10	0.506	0.760	0.886	0.946	0.975
	20	0.760	0.945	0.989	0.998	1.000
	30	0.885	0.989	0.999	1.000	1.000
	40	0.945	0.998	1.000	1.000	1.000
	50	0.975	1.000	1.000	1.000	1.000
0.3% (53 farms)	10	0.657	0.885	0.963	0.988	0.996
	20	0.885	0.988	0.999	1.000	1.000
	30	0.962	0.999	1.000	1.000	1.000
	40	0.988	1.000	1.000	1.000	1.000
	50	0.996	1.000	1.000	1.000	1.000

^1^ Farm-level prevalence in a population of 17,521 farms in a defined region (1,615,246 km^2^). ^2^ Farm-level sensitivity is the probability of a positive test on samples from an infected farm.

**Table 6 animals-14-00233-t006:** Cost per sampling per farm to collect samples and ship to laboratory ^1^.

Category	Cost per Item ^2^		No. Items	Cost per Sampling	
A. Sample collection. Assumes 10 pigs/farm/sampling					
Option 1. Serum samples					
Blood collection tubes (single-use)	0.525	0.563	10	5.26	5.64
Blood collection needles (single-use)	0.572	0.613	10	5.72	6.13
Plastic tube for pooling 5 samples ^3^	0.171	0.183	2	0.35	0.37
Disposable gloves	€0.064	$0.069	4 (2 pairs)	0.26	0.28
				€11.58	$12.42
Option 2. Swab samples (blood, nasal, oral, or fecal)					
Sample collection swabs	0.532	0.570	10	5.32	5.70
Transport medium, e.g., phosphate-buffered saline	0.036	0.039	5 mL	0.36	0.39
Plastic tube for pooling 5 samples	0.171	0.183	2	0.35	0.37
Disposable gloves	€0.064	$0.069	4 (2 pairs)	0.26	0.28
				€6.29	$6.74
B. Shipment of samples to the laboratory					
Insulated shipping container	5.167	5.540	1	5.17	5.54
Cold packs to ship with samples	6.529	7.000	1	6.53	7.00
Parcel shipping charge	€13.991	$15.000		13.99	15.00
				€25.69	$27.54

^1^ EUR (€) 1.00 = USD ($) 1.0721 U.S. (https://www.federalreserve.gov/releases/h10/current/ accessed on 19 June 2023). ^2^ Means of prices provided by three distributors in the U.S. ^3^ Cost analysis assumed blood samples would be centrifuged and serum pooled (5 pigs per pool) prior to shipment to avoid sample processing charges at the laboratory.

**Table 7 animals-14-00233-t007:** Cost of sampling and testing by specimen and test based on 3 test cost options.

		Serum Tested by PCR			Swabs Tested by PCR			Serum Tested by ELISA		
	Cost per test	EUR 18.65	EUR 23.32	EUR 27.98	EUR 18.65	EUR 23.32	EUR 27.98	EUR 4.66	EUR 7.00	EUR 9.33
Denominator		USD 20.00	USD 25.00	USD 30.00	USD 20.00	USD 25.00	USD 30.00	USD 5.00	USD 7.50	USD 10.00
Per farm in region ^1^		EUR 74.56	EUR 83.89	EUR 93.22	EUR 69.28	EUR 78.60	EUR 87.93	EUR 46.58	EUR 51.25	EUR 55.91
		USD 79.94	USD 89.94	USD 99.94	USD 74.27	USD 84.27	USD 94.27	USD 49.94	USD 54.94	USD 59.94
Per pig in region ^1^		EUR 0.025	EUR 0.029	EUR 0.032	EUR 0.023	EUR 0.027	EUR 0.030	EUR 0.016	EUR 0.018	EUR 0.019
		USD 0.027	USD 0.031	USD 0.034	USD 0.025	USD 0.029	USD 0.032	USD 0.017	USD 0.019	USD 0.020
Per pig in inventory Farms of ≤ 1000 pigs ^2^		EUR 0.240	EUR 0.270	EUR 0.299	EUR 0.223	EUR 0.253	EUR 0.283	EUR 0.150	EUR 0.165	EUR 0.180
		USD 0.257	USD 0.289	USD 0.321	USD 0.239	USD 0.271	USD 0.303	USD 0.161	USD 0.177	USD 0.193
Farms of 1001–4999 pigs ^3^		EUR 0.027	EUR 0.030	EUR 0.034	EUR 0.025	EUR 0.028	EUR 0.032	EUR 0.017	EUR 0.019	EUR 0.021
		USD 0.029	USD 0.032	USD 0.036	USD 0.027	USD 0.030	USD 0.034	USD 0.018	USD 0.020	USD 0.022
Farms of ≥ 5000 pigs ^4^		EUR 0.007	EUR 0.008	EUR 0.009	EUR 0.007	EUR 0.007	EUR 0.008	EUR 0.005	EUR 0.006	EUR 0.006
		USD 0.008	USD 0.009	USD 0.010	USD 0.007	USD 0.008	USD 0.009	USD 0.005	USD 0.005	USD 0.006

^1^ Estimates based on 17,521 farms in the region, holding 51,515,700 pigs. ^2^ Estimates based on 4422 farms with a mean inventory of 311 pigs. ^3^ Estimates based on 11,261 farms with a mean inventory of 2776 pigs. ^4^ Estimates based on 1838 farms with a mean inventory of 10,272 pigs.

## Data Availability

Current concentrated animal feeding operation (CAFO) permit data are available from the appropriate state authorities through the U.S. Freedom of Information Act.

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
