# Peer review of "Active Participatory Regional Surveillance for Notifiable Swine Pathogens"

_animals, 2024, doi:10.3390/ani14020233_

Round 1

Reviewer 1 Report

Comments and Suggestions for Authors

In the manuscript, the authors conducted a study of Active Participatory Regional Surveillance for Notifiable Swine Pathogens.

However, there are some comments.

1. Line 94-99. The purpose of the study is not formulated clearly! Write a complete objective according to the topic and content of the manuscript!

2. Line 143. The software used was developed or produced by whom. You must provide this information.

3. Line 488-497. The conclusion needs to be redone. Add the results of higher research to it. An analysis of the practical application of the results should also be provided.

Comments on the Quality of English Language

The manuscript is written in good English.

However, there are minor typos in the text.

Therefore, I recommend minor editing of the English language of the manuscript.

Author Response

We thank the Reviewers and note that their suggestions have greatly assisted in improving the manuscript.

Regards,

The Authors [Giovani Trevisan, Paul Morris, Gustavo S. Silva, Pormate Nakkirt, Chong Wang, Rodger Main, and Jeffrey Zimmerman]

REVIEWER 1

Comments and Suggestions for Authors

In the manuscript, the authors conducted a study of Active Participatory Regional Surveillance for Notifiable Swine Pathogens.  However, there are some comments.

  1. Line 94-99. The purpose of the study is not formulated clearly! Write a complete objective according to the topic and content of the manuscript!

We thank reviewer 1 for this excellent comment.  Lines 91 - 99 now reads,

The objective of this study was to characterize the performance of a surveillance design best described as, "collecting and testing a few targeted samples from each of many farms in a region".  In more formal terms, we analyzed the performance and cost of test-based, regional, active participatory surveillance based on targeted sampling of 10 poor-doing pigs by farm personnel (producer, staff, and/or veterinarian) followed by screening for the pathogen of interest in credentialed laboratories. Because our objective was to explore the feasibility and performance of this general design, the analysis did not include a sampling and testing process for a specific pathogen. However, a key assumption in the cost analysis was that 10 pigs would be sampled on each participating farm and the individual pig samples combined into two pools (5 pigs per pool) for antibody or nucleic acid testing.

  1. Line 143. The software used was developed or produced by whom. You must provide this information.

This information is provided in lines 113-116:  The Animal Disease Spread Model (ADSM) is public domain software designed to simulate the spread of infectious agents in livestock populations [22]. ADSM uses a static, fixed population and defines the population of animals at a single geographic location, i.e., a farm, as the epidemiological unit. 

Furthermore, the interested reader is directed to citation 22 for additional background:  Reeves et al., The North American Animal Disease Spread Model: a simulation model to assist decision making in evaluating animal disease incursions. Prev. Vet. Med. 2007, 82, 176-197. https://doi.org/10.1016/j.prevetmed.2007.05.019.

Finally, in lines 432-440 we note that:  The Animal Disease Spread Model (ADSM) [15] software used in Phase 1 provided substantial modeling flexibility and has previously been used to simulate the spread of ASFV in Vietnam [49], Aujeszky's disease virus in Thailand [53], CSFV in the Republic of Serbia [54], and PRRSV in both Uganda [55] and Canada [29].

  1. Line 488-497. The conclusion needs to be redone. Add the results of higher research to it. An analysis of the practical application of the results should also be provided.

We are uncertain what is meant by "redone" or "higher research".  There is certainly much more work to be done in this area, but we believe this initial study is complete unto itself and, indeed, exemplifies a practical application of the proposed approach. 

Comments on the Quality of English Language

The manuscript is written in good English.

However, there are minor typos in the text.

Therefore, I recommend minor editing of the English language of the manuscript.

The manuscript was scanned using Microsoft Word spelling and grammar checker using English (United States).  No errors were encountered.

Reviewer 2 Report

Comments and Suggestions for Authors

The paper presents an assessment of an active participatory design for regional surveillance of notifiable swine pathogens, wherein 10 samples collected by farm personnel from each participating farm are tested. The evaluation employs public domain software to simulate pathogen introduction and spread among 17,521 farms within a vast geographic region spanning 1,615,246 km². The study aims to gauge the efficacy of this surveillance design by estimating the probability of detecting at least one positive farm in the region based on varying percentages of participating farms (20%, 40%, 60%, 80%, 100%), farm-level detection probabilities (10%, 20%, 30%, 40%, 50%), and regional farm-level prevalence.

Results indicate that, at a 0.1% prevalence and a 30% farm-level detection probability, the participatory surveillance design achieved probabilities of 67%, 90%, and 97% for detecting at least one positive farm in the region with 20%, 40%, and 60% producer participation, respectively. The cost analysis, assuming pooling 10 individual pig samples into two samples for testing, reveals a range of €0.017 to €0.032 ($0.017 to $0.034 USD) per pig in the region, depending on specimen type (serum or swab) and test format (nucleic acid or antibody detection). Consequently, the study suggests that an active regional participatory surveillance design can effectively detect low-prevalence pathogens at a sustainable cost.

While the study attempts to assess the performance of this surveillance approach through simulation, several deficiencies and areas for improvement are noteworthy.

The paper lacks a comprehensive introduction that contextualizes the significance of regional surveillance for notifiable swine pathogens. A clearer connection between the research question and the broader context of animal health surveillance is needed.

The paper provides limited insight into the methodology, especially regarding the public domain software used for simulation. A more thorough explanation of the simulation model, its assumptions, and parameters would enhance the transparency and reproducibility of the study.

The limitations of the study, particularly in simulating real-world scenarios, are not sufficiently discussed.

The results are presented in a concise manner, but there is a lack of clarity in explaining the rationale behind the chosen prevalence rate (0.1%) and the farm-level detection probability (30%).

The paper suffers from occasional language issues and lack of clarity in expressions. Certain sentences are convoluted, making it challenging for the reader to follow the logical flow of the argument.

In conclusion, the paper has promising aspects, but it requires substantial improvements in contextualization, methodological transparency, discussion of limitations, results justification, cost analysis breakdown, and overall language clarity to enhance its scholarly value and impact.

Comments on the Quality of English Language

Improve English for clarity throughout the paper

Author Response

We thank the Reviewers and note that their suggestions have greatly assisted in improving the manuscript.

Regards,

The Authors [Giovani Trevisan, Paul Morris, Gustavo S. Silva, Pormate Nakkirt, Chong Wang, Rodger Main, and Jeffrey Zimmerman]

REVIEWER 2

Comments and Suggestions for Authors

The paper presents an assessment of an active participatory design for regional surveillance of notifiable swine pathogens, wherein 10 samples collected by farm personnel from each participating farm are tested. The evaluation employs public domain software to simulate pathogen introduction and spread among 17,521 farms within a vast geographic region spanning 1,615,246 km².  The study aims to gauge the efficacy of this surveillance design by estimating the probability of detecting at least one positive farm in the region based on varying percentages of participating farms (20%, 40%, 60%, 80%, 100%), farm-level detection probabilities (10%, 20%, 30%, 40%, 50%), and regional farm-level prevalence.

Results indicate that, at a 0.1% prevalence and a 30% farm-level detection probability, the participatory surveillance design achieved probabilities of 67%, 90%, and 97% for detecting at least one positive farm in the region with 20%, 40%, and 60% producer participation, respectively.  The cost analysis, assuming pooling 10 individual pig samples into two samples for testing, reveals a range of €0.017 to €0.032 ($0.017 to $0.034 USD) per pig in the region, depending on specimen type (serum or swab) and test format (nucleic acid or antibody detection). Consequently, the study suggests that an active regional participatory surveillance design can effectively detect low-prevalence pathogens at a sustainable cost.

While the study attempts to assess the performance of this surveillance approach through simulation, several deficiencies and areas for improvement are noteworthy.

The paper lacks a comprehensive introduction that contextualizes the significance of regional surveillance for notifiable swine pathogens.  A clearer connection between the research question and the broader context of animal health surveillance is needed.

The authors believe that the introduction clearly presents the problem.  Specifically, Paragraph 1 (lines 48-63) notes the changes in swine production that underlie the problem, i.e., fewer but larger farms and extensive movement of animals, transport, personnel, and product within and between regions.  Paragraph 2 continues by noting a few of our most egregious slip-ups, i.e., ASFV in Brazil, CSFV in the Netherlands, FMDV in the UK.  More examples could be added, but the point should be clear: in all of these cases, the major problem was a lapse in early detection of the pathogen, i.e., surveillance failed. 

The paper provides limited insight into the methodology, especially regarding the public domain software used for simulation. A more thorough explanation of the simulation model, its assumptions, and parameters would enhance the transparency and reproducibility of the study.

This information is provided in lines 113-116:  The Animal Disease Spread Model (ADSM) is public domain software designed to simulate the spread of infectious agents in livestock populations [22]. ADSM uses a static, fixed population and defines the population of animals at a single geographic location, i.e., a farm, as the epidemiological unit. 

Furthermore, the interested reader is directed to citation 22 for additional background:  Reeves et al., The North American Animal Disease Spread Model: a simulation model to assist decision making in evaluating animal disease incursions. Prev. Vet. Med. 2007, 82, 176-197. https://doi.org/10.1016/j.prevetmed.2007.05.019.

Finally, in lines 432-440 we note that:  The Animal Disease Spread Model (ADSM) [15] software used in Phase 1 provided substantial modeling flexibility and has previously been used to simulate the spread of ASFV in Vietnam [49], Aujeszky's disease virus in Thailand [53], CSFV in the Republic of Serbia [54], and PRRSV in both Uganda [55] and Canada [29].

The parameters used in the simulations are precisely described in Table 2.

Also note lines 432-440 in the Discussion:  The purpose of Phase 1 was to create datasets of swine farms of known infection status (negative, positive) for use in the detection simulations (Phase 2). The Animal Disease Spread Model (ADSM) [15] software used in Phase 1 provided substantial modeling flexibility and has previously been used to simulate the spread of ASFV in Vietnam [49], Aujeszky's disease virus in Thailand [53], CSFV in the Republic of Serbia [54], and PRRSV in both Uganda [55] and Canada [29]. To more fully explore the diversity of spread patterns, a total of 2,430 pathogen spread scenarios were simulated (Section 2.1.1) based on combinations of county level pig densities, index herd production types, and probabilities of transmission by direct contact, indirect contact, and area spread.

The limitations of the study, particularly in simulating real-world scenarios, are not sufficiently discussed.  The results are presented in a concise manner, but there is a lack of clarity in explaining the rationale behind the chosen prevalence rate (0.1%) and the farm-level detection probability (30%). 

The population of farms used in the simulations are real-world farms based on the permitting required to operate concentrated animal feeding operations.  The prevalence and detection levels noted by Reviewer 2 are only examples.  Table 5 and Figures 1 and 2 provide complete results over a range of values.

The paper suffers from occasional language issues and lack of clarity in expressions. Certain sentences are convoluted, making it challenging for the reader to follow the logical flow of the argument.

This comment was unexpected.  In the past, our writing has been well received.  Aggregating the authors' productivity, we have written > 300 refereed scientific publications in English, edited more than 6 books in English, and written > 20 book chapters in English.

In conclusion, the paper has promising aspects, but it requires substantial improvements in contextualization, methodological transparency, discussion of limitations, results justification, cost analysis breakdown, and overall language clarity to enhance its scholarly value and impact.

Comments on the Quality of English Language

Improve English for clarity throughout the paper 3

Already addressed (above).

Reviewer 3 Report

Comments and Suggestions for Authors

An overall very interesting simulation study using R statistical software  and ADSM as basic tools. A number of assumptions have been reasonably made (eg variable cost of shipping, and sampling efforts expected to be fully successful without requirement for second sampling of the same animal etc), thus results of cost/pig are very close to what is to be expected in real circumstances. A few improvements and reduction of repetitions observed in the discussions section are highlighted as comments in the attached pdf file.

Author Response

We thank the Reviewer 3 and note that their suggestions have greatly assisted in improving the manuscript.

Regards,

The Authors [Giovani Trevisan, Paul Morris, Gustavo S. Silva, Pormate Nakkirt, Chong Wang, Rodger Main, and Jeffrey Zimmerman]

REVIEWER 3

Comments and Suggestions for Authors

An overall very interesting simulation study using R statistical software and ADSM as basic tools.  A number of assumptions have been reasonably made (e.g., variable cost of shipping, and sampling efforts expected to be fully successful without requirement for second sampling of the same animal etc.), thus results of cost/pig are very close to what is to be expected in real circumstances.  A few improvements and reduction of repetitions observed in the discussions section are highlighted as comments in the attached pdf file.

Line 167-168 Please add as in previous parentheses.

Done as requested.

Line 221 Table 3 should be put after the start of the section not before in the final text.

We tried to accommodate this request but could not. Because of the size of the Tables and Figures, putting each of them where they should properly go (after they are introduced) created a domino effect of new problems = pages with large blank areas. Actually, we have spent an unjustifiable number of hours "typesetting" this document, i.e., trying to put it together as it should be.  This should be a function of the journal, not the researcher/author.

Line 227 Should be highlighted and discussed in the discussion even though it is a well- established route of transmission

A point well taken, but the focus of this research is on detection, not spread.  That is, the ADSM spread data are only the matrix in which to test the detection scheme.  In this paper, analyses of the spread parameters can only detract from the thrust of the message.  

Line 292 A million things could go wrong in this procedure of samples dispatch.

No doubt, things can and do go wrong.  Perhaps most of these errors are probably encountered and corrected at the point of sample receipt in the VDLs.  Regardless, this is the routine process by which sampling/testing is done in U.S. swine.  Actually, as things have evolved, most sampling is done by lay persons in the field.  In the larger view, it has worked pretty well.

Line 298 Price levels were based on typical cost levels in private labs?

Correct.  The costs used for ELISAs and PCRs are typical of those in U.S. laboratories.  The costs used for supplies were based on quotes from 3 distributors.       

Line 418 misuse

Thank you - corrected.

Line 421-423 Already suggested in L 119. Please omit.

Lines 436-447 have been explained previously. They do not add new information to the discussions section, thus they can be omitted, or reduced significantly.

Lines 460-466 are a repetition of previously described methods. Please see previous comments on the same page.

Addressing these three comments jointly:  we agree with you in principle.  However, as you know, the majority of researchers and academicians do not read.  Thus, the redundancy is intentional and designed to anticipate the fact that many (most?) will only skim the abstract and perhaps the discussion.  Nevertheless, it has been reduced somewhat, as requested.
